# Analysis of Interpolation Methods in the Validation of Backscattering Coefficient Products

**DOI:** 10.3390/s23010469

**Published:** 2023-01-01

**Authors:** Yanan Jiao, Fengli Zhang, Qiqi Huang, Xiaochen Liu, Lu Li

**Affiliations:** 1Aerospace Information Research Institute, Chinese Academy of Sciences, Beijing 100101, China; 2College of Resources and Environment, University of Chinese Academy of Sciences, Beijing 100049, China

**Keywords:** radar backscattering coefficient, validation of remote sensing products, FFT interpolation

## Abstract

Validation is the basis of synthetic aperture radar (SAR) image quantification applications. Based on the point target of the field site, the radiation characteristics of the backscattering coefficient image can be used to optimize the SAR imaging, and the product production system can be more closely targeted, to ensure the image product accuracy in the actual quantification application. In this study, the validation of the backscattering coefficient image was examined using calibrators, and the radiometric properties of the image were evaluated by extracting the radar cross-section of each point target. Bilinear interpolation and fast Fourier transform (FFT) interpolation methods were introduced for the local area interpolation of point targets, and the two methods were compared from the perspective of response function imaging and validation accuracy. The results show that the FFT interpolation method is more favorable for validating the backscattering coefficient.

## 1. Introduction

Satellite load calibration is the key to obtaining high-precision satellite products through which quantitative relationships between load observations and observed physical quantities can be established [1]. Validation uses ground truth values to assess the accuracy and uncertainty of remote sensing products and is an effective way to evaluate the quality, reliability, and applicability of remote sensing products [2,3]. Radiometric calibration is mainly used for single-look complex (SLC) data, which is based on reference targets with a known Radar Cross Section (RCS) to determine the calibration constant and thus obtain the backscattering coefficient value of the ground target. Nevertheless, in some cases only the backscattering coefficient product is available to the user, hence there is the need to evaluate the quality of the backscattering coefficient image. The validation is performed on the backscattering coefficient image using ground reference point targets to quantitatively evaluate the radiometric accuracy of the backscattering coefficient image which is very important for environmental parameter inversion and applications.

The backscattering coefficient refers to the radar scattering cross-section per unit irradiated area and is a measure of the result of the interaction of the incident electromagnetic wave with the ground target [4]. In addition to being related to radar system parameters, it mainly depends on the complex dielectric constant of the ground features, surface roughness, etc. [5]. The backscatter coefficient product is a synthetic aperture radar (SAR) load base radiation characteristic product, which is generally calculated from single-look complex (SLC) data using a calibration constant K [6]. The accuracy of the backscatter coefficient is the basis for carrying out the quantitative application of radar remote sensing, which usually uses calibrators deployed on the ground to extract the point target response energy from the image and compare it with the theoretical value to derive the accuracy of the image.

For point target response energy measurements, because point target peaks account for only a small number of pixels in the processed data, the target response within the peak neighborhood must be interpolated, and the sample window is expanded before analysis [7]. Prasad et al. [8] were the first to propose a fast Fourier transform (FFT) interpolation method and showed that the FFT method can significantly reduce the computational effort and estimate the intermediate values of discrete sequences within a very small error margin. Wang et al. [9] introduced common methods in SAR image interpolation, including linear interpolation, such as nearest-neighbor, bilinear, cubic convolution, FFT, and Sinc interpolation, which are commonly used in signal processing. Chen et al. [10], when using the peak method to measure the response energy, proposed that the FFT complementary zero method has a smaller root mean square error when interpolating Sinc curves compared to the cubic spline interpolation and will more closely fit the Sinc curves.

High-resolution SAR images are significantly affected by speckle noise, and different interpolation methods may significantly affect the extraction accuracy of the point target response energy. Therefore, this study explores the effects of different interpolation methods on the accuracy of the backscattering coefficient and provides suggestions on interpolation methods using experimental data obtained from the Common Application Support Platform of Land Observation Satellite for Country Civil Space Infrastructure (CASPLOS_CCSI) 1-m resolution C-SAR/01 Satellite Radiation Characterization Validation.

## 2. Methods

### 2.1. Backscattering Coefficient Product Validation Workflow

The process of validating the SAR image backscattering coefficient products is shown in Figure 1. If this is SLC data, the SLC data are processed to obtain the backscattering coefficient product using the SAR satellite calibration equation, calibration constant, incidence angle, and other related information. First, SAR-ground matching is undertaken. Affine transformation is performed on the latitude and longitude coordinates of the calibrators deployed in the field to obtain the row and column coordinates of the corresponding point target in the image. Subsequently, the local area around the point target is extracted and interpolated [11], and the RCS of the point target is calculated using the integration method [12]. Finally, this was compared with the theoretical RCS of the calibrator to calculate the authenticity check results.

A symmetric imaging window of $2^{k}\times 2^{k}$ size was selected, for which the point target position was the center. Because the point target occupies fewer image elements in the SAR image and is stored as discrete data, resulting in the loss of a certain degree of original signal information, the sample window must be extended via interpolation to recover the point target response.

After interpolation, the signal-to-clutter ratio (SCR) value is calculated for each point target using the method described in Equation (1). Point targets with a SCR higher than 20 dB [13] are considered valid and can be used for subsequent validation.
(1)$$SCR=\frac{\sigma_{i}^{p}\sin\theta }{\langle \sigma_{i}^{c}\rangle \delta_{a}\delta_{r}}$$
where $\sigma_{i}^{p}$ is the peak value of the backscattering coefficient in the energy integral region of the *i*-th point target; $\langle \sigma_{i}^{c}\rangle$ is the average backscattering coefficient value of the clutter background around the *i*-th point target; $\theta_{i}$ is the local incidence angle of the *i*-th point target; $\delta_{r}$ and $\delta_{a}$ are the range and azimuth pixel spacing respectively.

The measured values of RCS for the window integration method were calculated using Equation (2).
(2)$$RCS_{\mathrm{integral}}=\left(\sum_{i\in A}^{N_{A}}\sigma_{i}-\frac{N_{A}}{N_{B}}\sum_{i\in B}^{N_{B}}\sigma_{i}\right)\delta_{a}\delta_{r}$$
where $RCS_{\mathrm{integral}}$ is the measured radar scattering cross-sectional area of the point target based on the window integration, the unit is $m^{2}$; $\sigma_{i}$ is the calculated backscattering coefficient of the image element of the point target imaging window; *A* is the actual energy integration region of the point target, including the total number of image elements $N_{A}$; *B* is the background area, including the total number of image elements; and $N_{B}$, $\delta_{r}$ and $\delta_{a}$ are the range and azimuth pixel spacing respectively.

The necessary ground reference information, such as the theoretical RCS value of the corner reflector, can be calculated based on the shape and edge length of the corner reflector and waveband information of the satellite-based SAR [14]. The RCS theoretical value of the corner reflector was compared [15] to the actual value extracted from the image to generate the radiation validation index of the SAR image and evaluate the image characteristics.

In this study, the following indicators were extracted to analyze the image radiation characteristics.

(1)Relative radiation validation accuracy

Ideally, corner reflectors with the same nominal RCS value should have equal RCS measurements; relative radiation validation accuracy is obtained by statistical analysis of the measurements of multiple corner reflectors with the same nominal RCS. This indicator reflects the relative difference between measured values.
(3)$$\Delta_{R}=\sqrt{\sum_{i=1}^{N}\frac{{\left(RCS_{i}-\bar{RCS}\right)}^{2}}{N-1}}$$
where $\Delta_{R}$ is the relative radiation validation accuracy, and N is the number of point targets used for validation. $RCS_{i}$ is the RCS measurement value of the *i*-th point target, and $\bar{RCS}$ is the mean RCS measurement value of the point targets.

(2)Absolute radiation validation accuracy

The absolute radiometric validation explains the absolute difference between the measured and theoretical values. The difference between the measured RCS value of the point target in the backscattering coefficient image and the corresponding theoretical value was calculated, and the maximum value of the absolute value of the result was used as the result of the absolute radiation validation accuracy of the backscattering coefficient image.
(4)$$\Delta_{A}=Max\left(\left|RCS_{1}-RCS_{t_{1}}\right|,\left|RCS_{2}-RCS_{t_{2}}\right|,\cdots ,\left|RCS_{n}-RCS_{t_{n}}\right|\right)$$
where $\Delta_{A}$ is the absolute radiation validation accuracy; $RCS_{i}$ and $RCS_{t_{i}}$ are the RCS measurement and theoretical value of the target RCS at the *i*-th point, respectively.

### 2.2. Interpolation Methods

From the above analysis, it is clear that accurate interpolation is an important step in extracting the point target response energy for backscattering coefficient validation. Linear interpolation is weighted according to spatial proximity, and its error at the interpolation nodes is zero. It is simple and convenient compared to other interpolation methods [16]; therefore, it is widely used in remote sensing image processing. Among them, bilinear interpolation can realistically fit the original image, is computationally faster than cubic convolution [17], and is a common interpolation method in many image processing applications. However, the linear interpolation method only considers spatial proximity and ignores the fluctuating characteristics of the signal. By contrast, the FFT interpolation method increases the sampling frequency of the signal by complementing the high-frequency part of the frequency domain, which can make the signal wave smoother [18]. Therefore, in this study, bilinear and FFT interpolations were compared.

#### 2.2.1. Bilinear Interpolation

Bilinear interpolation, also known as quadratic linear interpolation, is the process of bilinearly interpolating a pixel point by using the four pixel values around it plus the distance weights to sum up the new interpolation result [19]; Figure 2 shows the schematic diagram of bilinear interpolation.

The value taken at the place of (x+u,y+v) is calculated using Formula (5):(5)$$\begin{array}{l} f(x+u,y+v)=(1-u)(1-v)f(x,y)+u(1-v)f(x+1,y)\\ \;+v(1-u)f(x,y+1)+uvf(x+1,y+1) \end{array}$$
where *x* and *y* are the row and column coordinates of the known points in the input image, respectively; *u* and *v* are the distance difference between the required point and the known point row and column coordinates respectively.

#### 2.2.2. FFT Interpolation

In digital image processing, an FFT zero-padding operation is often performed on the signal. The FFT interpolation method performs a Fourier transform on the signal in the time domain and then zero padding at a high frequency, that is, the middle part of the signal in the frequency domain, producing the effect of interpolation in the time domain. In Figure 3, (a) identifies the point target before interpolation, (b) shows the result after zeroing the spectrum, and (c) shows the point target after interpolation. The white cross-shaped dashed box in (b) identifies the invalid spectral region that can be zeroed, and the expanded point target can be obtained by inversion of the 2D discrete Fourier transform after zeroing.

The interpolation process based on the FFT is realized using one Fourier transform and one Fourier inverse transform, and its operation process is divided into three main steps. First, a fast Fourier transform is applied to the original sequence $X_{N}(k_{1},k_{2})$ of size N×N [17].
(6)$$X_{N}(k_{1},k_{2})=\sum_{n_{1}=0}^{N-1}\sum_{n_{2}=0}^{N-1}x(n_{1},n_{2})\exp\left\{-j2\pi (\frac{k_{1}n_{1}}{N}+\frac{k_{2}n_{2}}{N})\right\}\;\;\;\;n_{1},n_{2},k_{1},k_{2}\in \left[0,N-1\right]$$

Then, a new sequence $X_{M}(k_{1},k_{2})$ of size M×M is constructed by $X_{N}(k_{1},k_{2})$.
(7)$$X_{M}(k_{1},k_{2})=\left\{ \begin{array}{cc} L^{2}\times X_{N}(k_{1},k_{2}) & k_{1}\in [0,\frac{N-1}{2}],k_{2}\in [0,\frac{N-1}{2}]\\ 0 & k_{1}\in [\frac{N-1}{2}+1,M-\frac{N-1}{2}-1],k_{2}\in [0,\frac{N-1}{2}]\\ L^{2}\times X_{N}(k_{1}-M+N,k_{2}) & k_{1}\in [\frac{N-1}{2}+1,M-1],k_{2}\in [0,\frac{N-1}{2}]\\ 0 & k_{1}\in [0,M-1],k_{2}\in [\frac{N-1}{2}+1,M-\frac{N-1}{2}-1]\\ L^{2}\times X_{N}(k_{1},k_{2}) & k_{1}\in [0,\frac{N-1}{2}],k_{2}\in [\frac{N-1}{2}+1,M-1]\\ 0 & k_{1}\in [\frac{N-1}{2}+1,M-\frac{N-1}{2}-1],k_{2}\in [\frac{N-1}{2}+1,M-1]\\ L^{2}\times X_{N}(k_{1}-M+N,k_{2}) & k_{1}\in [\frac{N-1}{2}+1,M-1],k_{2}\in [\frac{N-1}{2}+1,M-1] \end{array}\right.$$

After that, the Fourier inversion of the $X_{M}(k_{1},k_{2})$ is constructed to get $\hat{x}(m,m)$.
(8)$$\hat{x}(m_{1},m_{2})=\frac{1}{M_{1}M_{2}}\sum_{k_{1}=0}^{M_{1}-1}\sum_{k_{2}=0}^{M_{2}-1}X_{M}(k_{1},k_{2})\exp\left\{+j2\pi \left(\frac{k_{1}m_{1}}{M}+\frac{k_{2}m_{2}}{M}\right)\right\}$$

In Equations (6)–(8), $x\left(n_{1},n_{2}\right)$ is the original signal sequence, $n_{1}$ and $n_{2}$ refer to the image coordinates of the original signal sequence. *N* refers to the row and column size of the original signal. *L* is the interpolation multiplier. *M* refers to the row and column size of the sequence after interpolation. $X_{N}(k_{1},k_{2})$ is the two-dimensional sequence after fast Fourier transform. $X_{M}(k_{1},k_{2})$ is the frequency domain sequence after zero-padding. $\hat{x}(m_{1},m_{2})$ is the interpolated signal sequence. $m_{1}$ and $m_{2}$ are the coordinates in the interpolated signal sequence.

The key to the FFT zero-padding interpolation method lies in the selection of the sub-window size and setting of the zero-padding multiplier. The response range of the calibrators on the image is different for different edge sizes; specifically, the sub-window needs to have the correct number of sampling points to ensure the effectiveness of the interpolation algorithm, but there should not be too many sampling points, otherwise it will affect computational efficiency [10].

## 3. Experimental Area and Data

This experiment, supported by CASPLOS_CCSI, relies on the Xilinhot SAR satellite calibration and validation. The surface of the field was mainly covered with kerchief fescue and sheep grass, and the terrain in the study area was flat and open. The scattering characteristics of the background features were uniform and stable without strong targets or electromagnetic interference, which can effectively reduce the interference of background clutter [20]. Seven corner reflectors were deployed for the experiment, and their distributions are shown in Figure 4.

The GaoFen-3 satellite [21] is an important Chinese SAR satellite, and it has played a significant role in applications within the fields of marine monitoring, disaster mitigation, environmental protection, water conservancy, agriculture and meteorology, etc. The C-SAR/01, launched on 23 November 2021, is the first follow-up satellite to GaoFen-3 and is designed to form part of the C-band SAR satellite constellation of China’s sea and land surveillance and monitoring system. The payload of C-SAR/01 is a C-band SAR with a center frequency of 5.4 GHz. It has 12 imaging modes such as spotlight, strip, and TOPSAR, which can acquire SAR images with a resolution of 1 m~500 m, swath of 10 km~650 km, and ranges from single polarization to full polarization to realize the monitoring of ocean and land resources. The imaging modes and parameters of C-SAR/01 are shown in Table 1.

The C-SAR/01 SAR data used in this study are listed in Table 2, and the data are Level-1A SLC images acquired on 12 May 2022. The imaging mode of the image is Ultra-Fine Strip (UFS) with a resolution of 3 m in both range and azimuth direction.

The results were obtained using Equation (9) to calculate the backscattering coefficient products in the experimental area [22], as shown in Figure 5.
(9)$$\sigma_{dB}^{0}=10\log_{10}\left(DN^{2}*{\left(\frac{QualifyValue}{32,767}\right)}^{2}*\sin\theta \right)-K_{dB}$$
where $\sigma_{dB}^{0}$ is the calculated backscattering coefficient value, and the unit is dB. DN is the digital number of the SAR image, $DN^{2}$ is obtained by summing the squares of the real and imaginary parts of the SLC data, i.e., $DN^{2}=I^{2}+Q^{2}$. QualifyValue and $K_{dB}$ are the image quantization maximum value and calibration constant respectively, which can be obtained from the image metadata file. θ is the local incidence angle, which can be obtained from the incidence angle file of C-SAR/01.

The seven corner reflectors used in this experiment were all developed for CASPLOS_CCSI, as shown in Figure 6. The inner leg length is 1 m. Plate curvature is less than ±2 mm. When the angle is adjusted, the deviation of azimuth angle and elevation angle is less than ±0.1°. Therefore, the corner reflectors used in the experiment can effectively guarantee RCS accuracy for radiometric validation.

## 4. Results and Discussion

According to Nyquist’s sampling theorem, if the signal is band-limited and the sampling frequency is higher than twice the signal bandwidth [23], the original continuous signal can be completely reconstructed from the sampled values. The SLC used in this experiment consists of a real part and imaginary part, which satisfies Nyquist’s sampling law [24]. Sinc interpolation [25] is a common method for complex signal processing [26,27] and can reconstruct as much continuous information as possible in discrete sampled images [28]. Therefore, for SLC data, the continuous information in the discrete sampled image can be reconstructed by Sinc interpolation.

However, in the backscattering coefficient, the pixel DN value only contains the amplitude information, which no longer satisfies the Nyquist sampling law, and thus the original information cannot be reconstructed by Sinc interpolation method. Therefore, we aim to find a suitable interpolation method, so that the information of the point target in the backscattering coefficient can be restored as much as possible to perform radiometric accuracy validation. Bilinear interpolation is a widely used conventional interpolation method in remote sensing image processing, simple but with high accuracy in most cases [29]. The FFT interpolation method is utilized for interpolating finite-time real sequences with the discrete Fourier transform. Compared to the classical interpolation method [30], the mean square error of FFT interpolation is significantly lower. Besides, the FFT interpolation method requires less computation and has a higher accuracy for long sequence data [8].

In order to evaluate the efficiency of bilinear and FFT interpolation methods, Sinc interpolation on the SLC data was used as a reference. In this experiment, after the backscattering coefficient data to be examined were located at the point target, based on the position information of the calibrator, a local area of 32 × 32 pixels around the corner reflector was selected, and the backscattering coefficient data within the sample window were subjected to 8-fold bilinear interpolation and FFT interpolation, respectively, and the two methods were compared.

### 4.1. Point Target Response Function Analysis

For the results obtained by the two interpolation methods, the profile lines were extracted along the range and azimuth directions of the peak points and compared with the Sinc interpolation results of the SLC data; the results of CR-1 are shown in Figure 7.

Observing the range and azimuth profile lines in the above figure, the image as a whole shows the same trend, but the results obtained by the bilinear interpolation method present jaggedness, which do not fit the Sinc interpolation results of the SLC image well, which is a defect of the algorithm itself. The discontinuity of the data may introduce errors in the window integration method for RCS calculation, while the FFT interpolation method is smoother in fitting the results, so the effect of the FFT interpolation method is better than the bilinear interpolation from the point target response function profile line.

The two-dimensional interpolation results for the (a) backscattering coefficient products, (b) Sinc interpolation of SLC data, (c) bilinear interpolation, and (d) FFT interpolation are shown in Figure 8.

Visually, the point target behaves as a sine-type function in the processed backscattering coefficient image [7], and the image is symmetric and shows fluctuations. Both the FFT interpolation and bilinear interpolation results of the backscattering coefficient are similar to the Sinc interpolation results of the SLC data, but the FFT interpolation results are smoother and better reflect the characteristics of the fluctuation of the point target signal itself, complementing some of the information loss due to resolution limitations. On the other hand, the bilinear interpolation results are calculated only from the perspective of spatial proximity, and although they fit the original backward scattering coefficient data more closely, the results are more rigid and lose some signal information, which may affect the RCS extraction accuracy.

### 4.2. RCS Extraction Results

The interpolation multiples affect the sampling resolution and spectral resolution, as well as the radiometric validation accuracy [31]. After comparing the validation accuracy indicators with different interpolation multiples, it shows the validation accuracy is the highest at eight times interpolation. Therefore, in this study, an eight-fold interpolation is uniformly performed on the local area of the point target. The point target SCR and RCS extracted after the Sinc interpolation of the SLC data, and the point target RCS extracted after bilinear and FFT interpolation of the backscattering coefficient are listed in Table 3.

From the graphical results, the FFT interpolation of the backscattering coefficient is closer to the theoretical value of the RCS, and the standard deviation is smaller. Corner reflectors appear as point targets on SAR images. According to Equations (6)–(8), FFT interpolation increases the sampling frequency by complementing zeros in the high frequency portion of the frequency domain, resulting in the signal being interpolated in the time domain [8]. Nevertheless, bilinear interpolation is based on the spatial proximity of the imaging, solving for unknown points based on adjacent known points. As can be seen from Figure 7 and Figure 8, the FFT interpolation method provides a better fit to the fluctuating characteristics of the original signal, and the FFT interpolation method can better restore the signal characteristics of point targets than bilinear interpolation. When the signal recovery in the local area of the point target is greater, the calculated RCS is more accurate. Therefore, FFT is more suitable for the interpolation of point target signals and thus more suitable for backscattering coefficient validation. The RCS values of each corner reflector extracted using Sinc, Bilinear and FFT interpolation methods are shown in Figure 9.

To compare the effects of different interpolation methods on the validation accuracy, the relative validation accuracy and absolute validation accuracy were extracted for Sinc, Bilinear and FFT interpolation methods according to Equations (3) and (4) and listed in Table 4.

By analyzing and comparing the parameters listed in the table, the following conclusions can be drawn: (1) From the relative validation accuracy shown in the table, it can be seen that the results obtained by both FFT interpolation and bilinear interpolation methods are stable, but the FFT interpolation method is slightly better. (2) The absolute validation accuracy is the maximum of all point target errors in each method, and the RCS errors extracted after FFT interpolation are smaller. In summary, the FFT interpolation method has somewhat higher accuracy in the backscattering coefficient validation.

## 5. Conclusions

Validation is a key step in the quantitative application of the SAR backscattering coefficient products. This paper briefly describes the backscattering coefficient validation process, combines the measured data, extracts the local area of the point target in the C-SAR/01 image for interpolation, calculates the radar scattering cross-sectional area of the point target using the integral method, and generates validation indexes. The interpolation effects of bilinear interpolation and FFT interpolation were compared using the response function and validation accuracy after interpolation.

From the point target response function, using the FFT interpolation method to interpolate the local signals around the calibrators, the interpolation result profile lines were more consistent with the point target characteristics, retaining the original features of the image, and the range and azimuth profile curves were smoother. In contrast, bilinear interpolation results in a jagged image that may cause the loss of some signals. Among the validation indices calculated after using different interpolation methods, the RCS of the calibrators obtained by the FFT interpolation method is closer to the theoretical value and more stable among individual point data, which truly reflects the radiation characteristics of the image. In subsequent studies, in-depth research can be conducted on how to suppress the ringing effect of the interpolated image by adding windows according to the FFT interpolation principle and reducing the influence of background clutter on RCS extraction to improve validation accuracy.

## Figures and Tables

**Figure 1 sensors-23-00469-f001:**
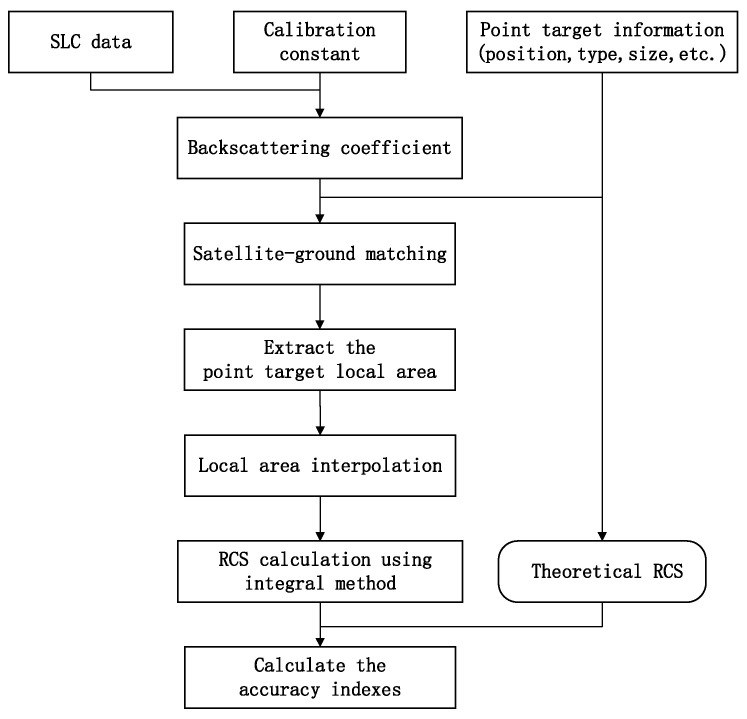
The validation process for the backscattering coefficient.

**Figure 2 sensors-23-00469-f002:**
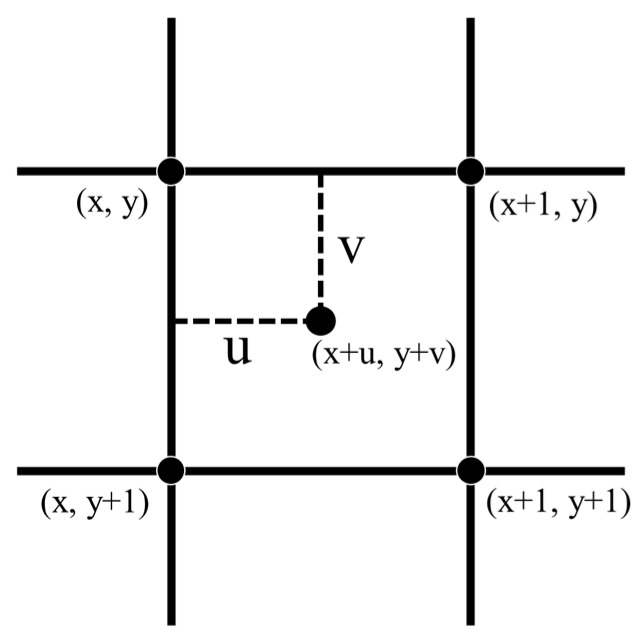
Schematic diagram of the bilinear interpolation method.

**Figure 3 sensors-23-00469-f003:**
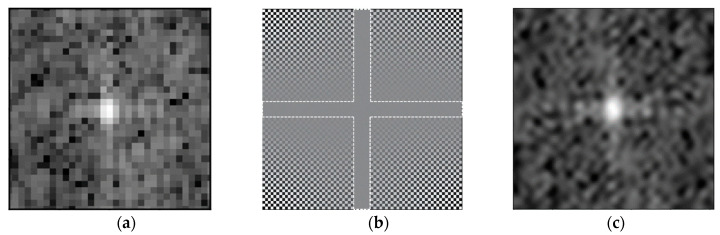
Schematic diagram of FFT interpolation. (**a**) Point target before interpolation. (**b**) Frequency graph after zero padding. (**c**) Point target after interpolation.

**Figure 4 sensors-23-00469-f004:**
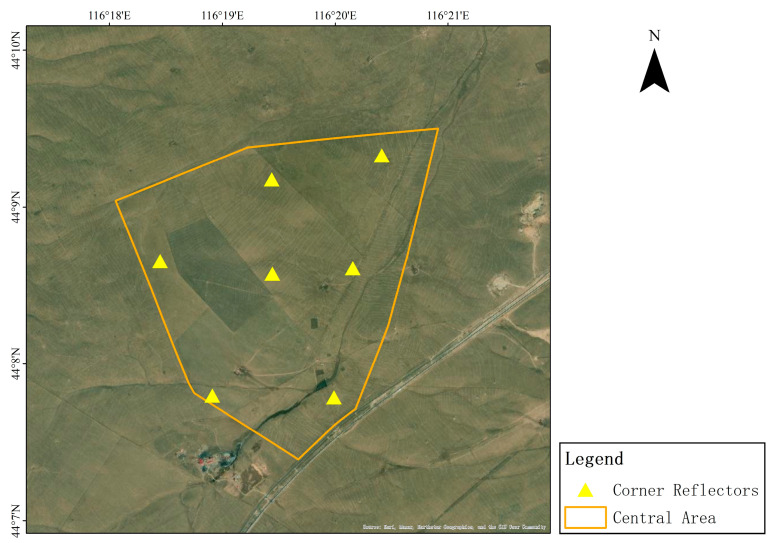
Study area coverage.

**Figure 5 sensors-23-00469-f005:**
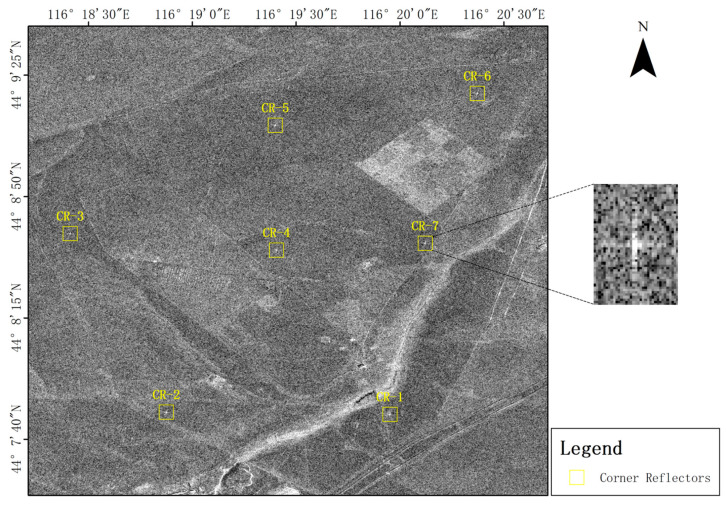
Geocoded backscattering coefficient products.

**Figure 6 sensors-23-00469-f006:**
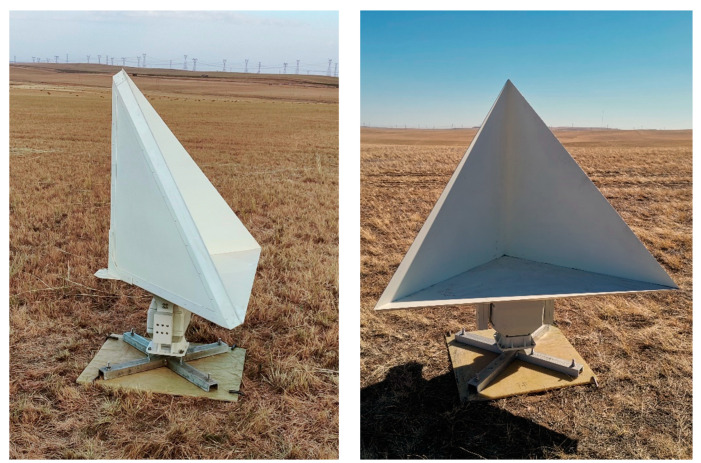
Automatic trihedral corner reflectors.

**Figure 7 sensors-23-00469-f007:**
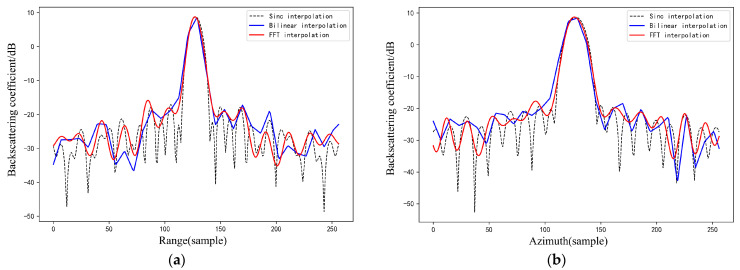
Interpolation results along range and azimuth direction using Sinc, Bilinear and FFT methods for point target (taking CR-1 as an example). (**a**) Interpolation results along range direction. (**b**) Interpolation results along azimuth direction.

**Figure 8 sensors-23-00469-f008:**
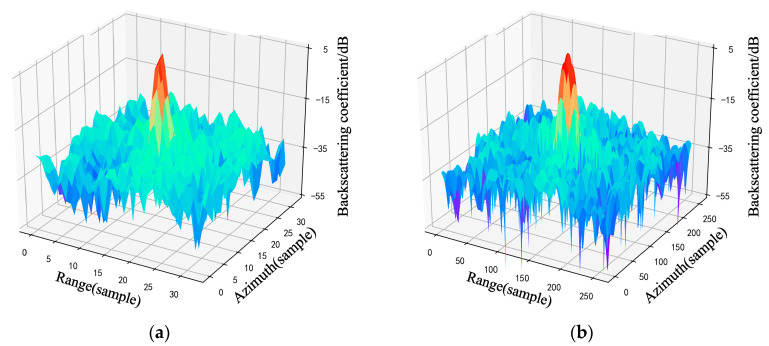
The backscattering coefficient image of CR-1 and the two-dimensional interpolation results using Sinc, Bilinear and FFT method. (**a**) Raw backscattering coefficient data; (**b**) Sinc interpolation results of SLC data; (**c**) Bilinear interpolation results of the backscattering coefficient; (**d**) FFT interpolation results of the backscattering coefficient.

**Figure 9 sensors-23-00469-f009:**
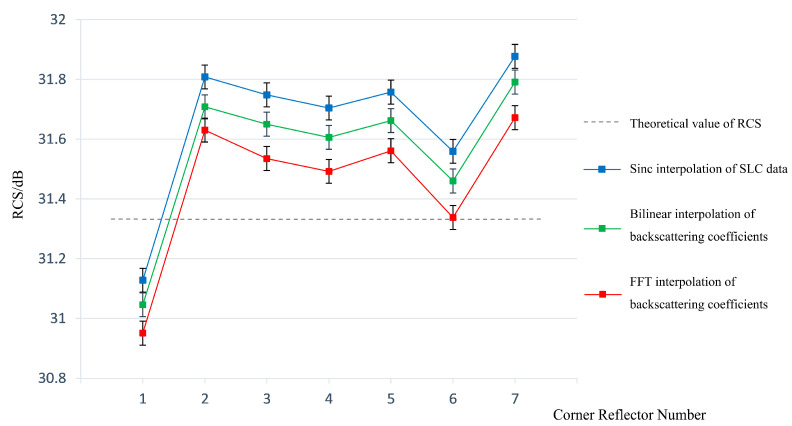
The extracted RCS values of each corner reflector using Sinc, Bilinear and FFT interpolation methods.

**Table 1 sensors-23-00469-t001:** Imaging modes and parameters of C-SAR/01.

Imaging Mode		Nominal Resolution/m	Swath Width/km	Polarization
Spotlight (SL)		1	10	HH/VV
Stripmap	Ultra-Fine Strip (UFS)	3	30	HH/VV
	Fine Strip I (FSI)	5	50	HH + HV/VV + VH
	Fine Strip II (FSII)	10	100	HH + HV/VV + VH
	Standard Strip (SS)	25	130	HH + HV/VV + VH
	Quad Polarization Strip I (QPSI)	8	30	Quad-polarization
	Quad Polarization Strip II (QPSII)	25	40	Quad-polarization
Scan	Narrow ScanSAR (NSC)	50	300	HH + HV/VV + VH
	Wide ScanSAR (WSC)	100	500	HH + HV/VV + VH
	Global (GLO)	500	650	HH + HV/VV + VH
Wave imaging (WAV)		8	20	Quad-polarization
Expanded incidence angle (EXT)	Low Incidence	25	130	HH + HV/VV + VH
	High Incidence	25	80	HH + HV/VV + VH

**Table 2 sensors-23-00469-t002:** Imaging parameters of C-SAR/01 image.

Date	12 May 2022
Ascending or Descending	Descending
Look Direction	Right
Incidence Angle	28.43–30.57°
Imaging Mode	UFS
Polarization Mode	HH
Range Resolution/m	3
Azimuth Resolution/m	3
Range pixel space/m	1.124222
Azimuth pixel space/m	1.669818

**Table 3 sensors-23-00469-t003:** SCR and RCS calculated for different interpolation methods.

CR Number	SCR/dB	Theoretical Value/dB	Sinc Interpolation of SLC Data/dB	Backscattering Coefficient/dB	
				Bilinear Interpolation	FFT Interpolation
CR-1	36.903	31.332	31.128	31.046	30.951
CR-2	38.181	31.332	31.808	31.708	31.63
CR-3	38.262	31.332	31.748	31.65	31.535
CR-4	37.398	31.332	31.704	31.606	31.492
CR-5	38.572	31.332	31.757	31.662	31.561
CR-6	37.678	31.332	31.559	31.46	31.338
CR-7	39.905	31.332	31.877	31.791	31.672

**Table 4 sensors-23-00469-t004:** Validation accuracy of different interpolation results.

	Relative Validation Accuracy/dB	Absolute Validation Accuracy/dB
**Sinc interpolation of SLC data**	0.252	0.544
**Bilinear interpolation of backscattering coefficient**	0.23	0.459
**FFT interpolation of backscattering coefficient**	0.228	0.381

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
