# Peer review of "Analysis of Interpolation Methods in the Validation of Backscattering Coefficient Products"

_sensors, 2023, doi:10.3390/s23010469_

Round 1

Reviewer 2 Report

1. In Figure 1, the words in this element differ in terms of upper and lower case.

2. Can the authors add some introductions about C-SAR/01?

3. In line 163, some descriptions may be not clear, “it will affect computational accuracy” may be revised as “it will affect computational efficiency”.

4. In line 165, Abbreviation CASPLOS_CCSI should be introduced with full name where they first appear in the paper.

5. In lines 174-176. More information should be added. For example, what’s the meaning of ultrafine? It may be known to the readers familiar with C-SAR/01, however, many may be confused about this.

6. In line 181, equation (8). The physical meaning represented by each symbol should be explained.

7. In Figure 5, it seems that the the image has poor image quality and low SNR. Can you give some explanation?

8. The detailed information on corner reflectors in Figure 6 should be added in the paper.

9. In Figure 7. Why first introduce (b) and then (a)? Another problem is that the horizontal and vertical coordinates are not marked in the figure.

10. Figure 8 and Figure 9, have the same problems as Figure 7.

11. In Table 2 and Table 3, please state the unit.

12. Although the RCS errors extracted after the FFT interpolation are smaller, the authors didn’t explain the reasons, please add some theoretical analysis.

13. The references should be modified as per standard format. For example, in reference 1, why “satellite” is in lowercase?

Round 2

Reviewer 2 Report

Thanks for the author's reply. I have only two questions left.

1. ’In figure 7 and figure 8, "backscattering coefficient" should be "Backscattering coefficient:

2. In the introduction of Gaofen-3, the following reference should be added to the paper.

Sun, Jili, Yu, Weidong, Deng, Yunkai. The SAR Payload Design and Performance for the GF-3 Mission. SENSORS.
